# Effects of Recessed-Gate Structure on AlGaN/GaN-on-SiC MIS-HEMTs with Thin AlO_x_N_y_ MIS Gate

**DOI:** 10.3390/ma13071538

**Published:** 2020-03-27

**Authors:** Hyun-Seop Kim, Myoung-Jin Kang, Jeong Jin Kim, Kwang-Seok Seo, Ho-Young Cha

**Affiliations:** 1School of Electronic and Electrical Engineering, Hongik University, Seoul 04066, Korea; hyunseop0426@mail.hongik.ac.kr; 2Department of Electrical and Computer Engineering, Seoul National University, Seoul 08826, Korea; mjkang5@snu.ac.kr (M.-J.K.); ksseo@snu.ac.kr (K.-S.S.); 3Metamaterial Electronic Device Research Center, Hongik University, Seoul 04066, Korea; jeongjin19@hongik.ac.kr

**Keywords:** aluminum oxynitride, GaN-on-SiC, metal-insulator-semiconductor, high electron mobility transistor, gate recess, leakage current, Johnson’s figure of merit (J-FOM)

## Abstract

This study investigated the effects of a thin aluminum oxynitride (AlO_x_N_y_) gate insulator on the electrical characteristics of AlGaN/GaN-on-SiC metal-insulator-semiconductor high electron mobility transistors (MIS-HEMTs). The fabricated AlGaN/GaN-on-SiC MIS-HEMTs exhibited a significant reduction in gate leakage and off-state drain currents in comparison with the conventional Schottky-gate HEMTs, thus enhancing the breakdown voltage. The effects of gate recess were also investigated while using recessed MIS-HEMT configuration. The Johnson’s figures of merit (= f_T_ × BV_gd_) for the fabricated MIS-HEMTs were found to be in the range of 5.57 to 10.76 THz·V, which is the state-of-the-art values for GaN-based HEMTs without a field plate. Various characterization methods were used to investigate the quality of the MIS and the recessed MIS interface.

## 1. Introduction

AlGaN/GaN high-electron-mobility transistors (HEMTs) have been developed for high-power and high-frequency applications because of their excellent properties, such as high electron mobility of two-dimensional electron gas (2-DEG) channel, high breakdown field, and wide energy bandgap [1,2,3]. Although AlGaN/GaN HEMTs have high commercial applications in high-frequency power amplifier industries, the conventional Schottky-gate HEMTs suffer from large gate leakage current [4], which limits the output power performance and induces reliability issues [5]. Therefore, AlGaN/GaN metal-insulator-semiconductor-HEMTs (MIS-HEMTs) that employ MIS gate insulators would be good alternatives for reducing the gate leakage current [6,7] and increasing the breakdown voltage [8,9]. Superior DC and RF characteristics in comparison with the conventional Schottky-gate HEMTs have been reported by using various insulators for AlGaN/GaN MIS-HEMTs, such as SiO_2_ [10], SiN_x_ [11], Al_2_O_3_ [12], HfO_2_ [13], and Sc_2_O_3_ [14], etc. However, a drawback of the AlGaN/GaN MIS-HEMT is the low transconductance (g_m_) that is caused by the increase in the effective barrier thickness [15,16], which limits the RF performance [17]. A possible solution can be achieved by the use of a recessed AlGaN/GaN MIS-HEMT structure in conjunction with a thin gate insulator, where the gate region was recessed to enhance the g_m_ [18,19]. However, the gate recess configuration results in a reduced 2-DEG channel concentration and, thus, limits the output current density and RF performance. Therefore, the trade-off relationship must be carefully considered while employing the recessed MIS gate structure.

In this study, we have investigated the effects of the gate recess depth on the DC and RF characteristics of recessed AlGaN/GaN MIS-HEMTs with an aluminum oxynitride (AlO_x_N_y_) MIS gate; the performance was evaluated using Johnson’s figure of merit (J-FOM = f_T_ × BV_gd_), which is the norm for evaluating high-frequency power transistors [20,21]. AlO_x_N_y_ films can be deposited while using various deposition techniques, among which plasma enhanced atomic layer deposition (PEALD) is a good choice for producing high quality thin films [22]. While a few studies on AlO_x_N_y_ gate insulators have been reported for enhancement-mode power switching devices [22,23,24], no in-depth study has been reported for a recessed AlGaN/GaN RF-HEMT with a thin AlO_x_N_y_ gate insulator.

## 2. Experimental Methods

The AlGaN/GaN-on-SiC wafer was procured from a commercial wafer vendor. The epitaxial layers consisted of a 25 nm unintentionally doped Al_x_Ga_1−x_N (x = 0.255) barrier layer, a 2 μm GaN buffer layer with a Fe-doped region away from the channel, and a nucleation layer on a 500 μm 4H-SiC semi-insulating substrate. The device fabrication was initiated with solvent cleaning, which was followed by a diluted HF (10:1) treatment for 5 min. for the removal of surface contaminants and native oxides. After the cleaning of the wafer, a 100 nm SiN_x_ film was pre-passivated while using the catalytic chemical vapor deposition (Cat-CVD) method to protect the GaN surface during the high-temperature ohmic annealing [25]. Subsequently, a Si/Ti/Al/Mo/Au (= 5/20/80/35/50 nm) ohmic metal stack was evaporated and annealed at 800 °C for 1 min. in N_2_ ambient. Mesa isolation was carried out using low-damage Cl_2_/BCl_3_-based inductively coupled plasma reactive ion etching (ICP-RIE). The mesa isolation depth was ~550 nm. The pre-passivated SiN_x_ layer was removed using a low damage dry etching process with SF_6_ plasma in order to remove the damaged interface between the pre-passivation layer and AlGaN surface [26]. A new 140 nm SiN_x_ passivation layer was subsequently re-deposited using Cat-CVD. After the patterning of the gate region, the SiN_x_ passivation layer from the gate foot region was etched away using low-damage SF_6_ plasma etching. Subsequently, the AlGaN layer under the gate region was recessed while using a low-power Cl_2_/BCl_3_-based ICP-RIE. After the completion of the gate recess process, a 5 nm AlO_x_N_y_ thin film was deposited as a gate insulator using a PEALD system. The detailed optimization of the PEALD AlO_x_N_y_ film was previously reported in Ref. [22]. According to x-ray photoelectron spectroscopy analysis, the nitrogen incorporation in AlO_x_N_y_ was ~4%. The gate electrode was formed by the Ni/Mo/Au (= 40/15/400 nm) evaporation. Non-recess MIS-gate and Schottky-gate HEMTs were also fabricated for comparison. The various gate schemes that were compared in this study were Schottky-gate, non-recess MIS, 7.5 nm recess MIS (i.e., with a remaining AlGaN barrier thickness of 17.5 nm), and 11 nm recess MIS (i.e., with a remaining AlGaN barrier thickness of 14 nm) structures. The fabricated devices had a source-to-gate distance of 1.5 μm, a gate foot length of 0.5 μm, and a gate-to-drain distance of 3.5 μm. Figure 1 shows the cross-sectional schematics of the fabricated devices.

## 3. Results and Discussion

The transfer current-voltage (I-V) characteristics of the fabricated devices were measured at V_ds_ = 10 V, during which forward and reverse direction sweeps were carried out to investigate the hysteresis characteristics. Negligible hysteresis was observed for all devices, except the 11 nm recess MIS device, as shown in Figure 2. The fabricated Schottky-gate device exhibited a maximum transconductance (g_m.max_) of 231 mS/mm with the gate leakage and an off-state drain leakage current density of ~2 × 10^−4^ A/mm. The non-recess MIS structure exhibited a g_m.max_ of 211 mS/mm with the gate leakage and an off-state drain leakage current density of ~10^−6^ A/mm. Although the g_m.max_ of the non-recess MIS device was slightly lower than that of the Schottky-gate device due to an increase in the effective barrier thickness [15,16], the gate leakage current was dramatically reduced by two orders of magnitude by employing the MIS gate. In addition, the recessed MIS devices exhibited superior characteristics in both leakage current and g_m.max_, in comparison with the Schottky and non-recess MIS gate devices. The 7.5 and 11 nm recess MIS devices had leakage current densities of ~3 × 10^−7^ A/mm and ~10^−10^ A/mm, respectively. The improvement in g_m.max_ was caused due to the reduced effective barrier thickness. While the Schottky forward turn-on characteristics limit the maximum positive gate voltage range for the Schottky-gate device, the MIS and the recessed MIS gate configurations allow for a higher positive gate voltage range due to the insulating AlO_x_N_y_ blocking layer. For example, the forward gate breakdown characteristics that were measured for a 11 nm recess MIS-gate structure exhibited a forward gate breakdown voltage of 3.35 V, which corresponded to a breakdown field of ~6.6 MV/cm (see Figure 3).

Figure 4 illustrates the output characteristics of the fabricated devices. It should be noted that the gate voltage ranges from −7 to 3 V for the MIS gate devices, while it ranges from −7 to 1 V for the Schottky-gate device because of the forward Schottky turn-on problem [4]. The Schottky-gate, non-recess MIS, 7.5 nm recess MIS, and 11 nm recess MIS devices exhibited maximum drain currents of 939, 1075, 1023, and 916 mA/mm, respectively. The higher voltage range of the MIS devices resulted in a higher maximum drain current density. All of the devices exhibited excellent drain current saturation and pinch-off characteristics.

The pulsed I-V characteristics for the different devices were measured by employing different quiescent drain bias voltages in order to investigate the current collapse characteristics, as shown in Figure 5. The quiescent gate bias voltages were determined by taking account of deep pinch-off conditions. The pulse width was 200 ns with a period of 1 ms. All of the devices, which were associated with a high-quality passivation process, exhibited excellent pulsed I-V characteristics without an additional field plate. The non-recess MIS and the 7.5 nm recess MIS devices exhibited slightly better characteristics in comparison with the Schottky-gate device, which elucidates that the additional AlO_x_N_y_ film was effective in further suppressing the surface trapping effects. It was assumed that a relatively worse pulsed characteristic of the 11 nm recess MIS device was found due to the presence of a very thin reminiscent AlGaN barrier, thus causing a more sensitive response to any surface trapping effects.

Figure 6 compares the off-state breakdown characteristics of the fabricated devices measured under the deep pinch-off conditions. For the conventional Schottky-gate device, the breakdown voltage defined at 1 mA/mm was found to be 46 V with high off-state leakage current. In contrast, the non-recess MIS device exhibited a breakdown voltage of 298 V with the suppressed leakage current in comparison with the Schottky-gate device. The recessed MIS gate devices achieved further enhancements in breakdown voltage and a significant reduction in leakage current. It is speculated that the leakage current that was observed in this work was mainly induced near the AlGaN surface or passivation interface, because the leakage current level decreased with increasing the recess depth. The 7.5 nm recess MIS and 11 nm recess MIS devices exhibited breakdown voltages of 496 and 646 V, respectively. It was evident that the improved breakdown characteristics of the recessed MIS devices were associated with the decrease in the leakage current along with the reduced electric field due to the voltage drop across the gate insulator.

Capacitance-voltage (C-V) measurements were carried out for Schottky-gate and MIS-gate devices in order to investigate the effective gate capacitance and the interface quality of the MIS devices. Figure 7 compares the measured C-V characteristics, where negligible hysteresis was observed, regardless of the device type, which means that the interface conditions of the MIS and recessed MIS devices are as good as the Schottky-gate devices. The effective capacitance can be expressed by employing a series connection of the AlGaN barrier and AlO_x_N_y_ insulator, from which the extracted dielectric constant of the AlO_x_N_y_ film was 8.4. The interface trap density was extracted by a conductance method [27], where the C-V measurements were carried out from 1 kHz to 1 MHz, in order to further evaluate the MIS interface conditions. The extracted interface trap density was < 10^11^ cm^−2^·eV^−1^ at the trap energy level of 0.4 eV from the conduction band edge, which indicates excellent MIS interface quality, as shown in Figure 8.

The small-signal characteristics were measured for the fabricated devices with a channel width of 2 × 50 μm. The S-parameter for each device was measured from 100 MHz to 40 GHz at the drain voltage of 28 V, and the gate bias point was set at the maximum g_m_ point. The measured cut-off frequency (f_T_) and the maximum oscillation frequency (f_MAX_) were extracted from the current gain (|H_21_|) and the maximum stable/available gain (MSG/MAG), respectively, as shown in Figure 9. The Schottky-gate device exhibited f_T_ of 18.9 GHz and an f_MAX_ of 60 GHz, whereas the non-recess MIS device exhibited f_T_ of 18.7 GHz and f_MAX_ of 54 GHz. The highest f_T_ of 21.7 GHz was achieved by the 7.5 nm recess MIS structure. On the other hand, the 11 nm recess MIS device exhibited the lowest f_T_ of 13.7 GHz. The gate capacitance was measured for different devices in order to analyze the behavior of f_T_ as a function of the gate structure. The gate capacitance varied widely for different gate structures, as shown in Figure 7. The degradation in f_T_ for the 11 nm recess MIS device can be explained by the high gate capacitance value that is caused by the very thin effective barrier thickness. This is a trade-off relationship for f_T_ between g_m_ and the gate capacitance [28]. The degradation of f_MAX_ for recessed MIS devices is associated with increased feedback capacitance [28]. An additional source field plate process can be used to reduce the feedback capacitance and mitigate the degradation of f_MAX_. 

Figure 10 shows the comparison of J-FOM (= f_T_ × BV_gd_) as a function of the gate length for various GaN-based HEMTs [20,21,29,30,31,32,33,34,35,36,37,38,39,40,41,42,43]. The MIS-HEMTs that were fabricated in this study exhibited the J-FOM in the range between 5.57 and 10.67 THz·V, which are the state-of-the-art values reported for GaN-based HEMTs. It is suggested that the thin AlO_x_N_y_ layer is an excellent gate insulator for reducing the gate leakage current and enhancing the breakdown voltage of AlGaN/GaN HEMT.

## 4. Conclusions

A 5 nm AlO_x_N_y_ thin film was investigated for its use as a gate insulator for AlGaN/GaN-on-SiC HEMT. A high-quality PEALD AlO_x_N_y_ deposition process was utilized for device fabrication where the gate structure was varied to be Schottky-gate, MIS-gate, and recessed MIS-gates. The gate leakage current and the breakdown voltage were significantly improved by employing a thin AlO_x_N_y_ MIS-gate configuration, which led to a significant improvement in the J-FOM values. The AlOxNy MIS-HEMTs attained state-of-the-art J-FOM values in the range of 5.57 to 10.76 THz·V. Moreover, by employing the pulsed and C-V characteristics, it was confirmed that the AlO_x_N_y_ MIS had excellent interface conditions. The maximum f_T_ was achieved while using a shallow recess depth due to the trade-off relationship between g_m_ and gate capacitance; thus, the gate recess depth must be carefully optimized.

## Figures and Tables

**Figure 1 materials-13-01538-f001:**
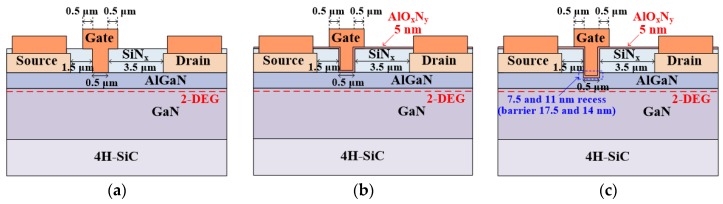
Cross-sectional schematics of the AlGaN/GaN high-electron-mobility transistors (HEMTs): (**a**) Schottky-gate, (**b**) non-recess metal-insulator-semiconductor (MIS), and (**c**) recess MIS structures.

**Figure 2 materials-13-01538-f002:**
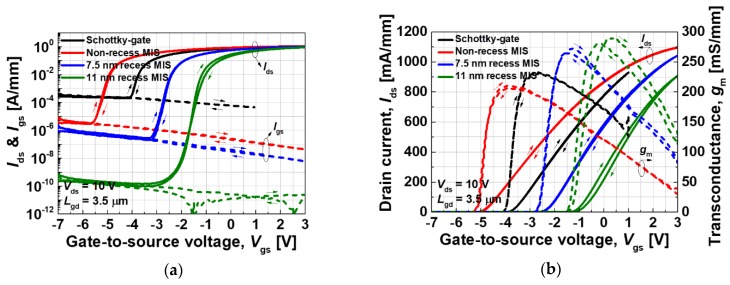
Comparison of the transfer characteristics; (**a**) drain and gate current characteristics in a logarithmic scale and (**b**) transconductance characteristics.

**Figure 3 materials-13-01538-f003:**
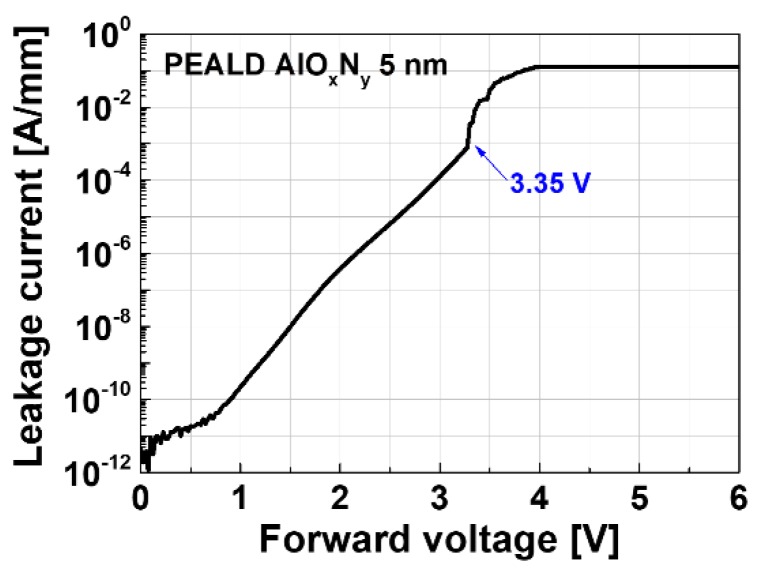
Forward breakdown characteristics of a 5 nm AlO_x_N_y_ film on an 11 nm recessed AlGaN/GaN surface.

**Figure 4 materials-13-01538-f004:**
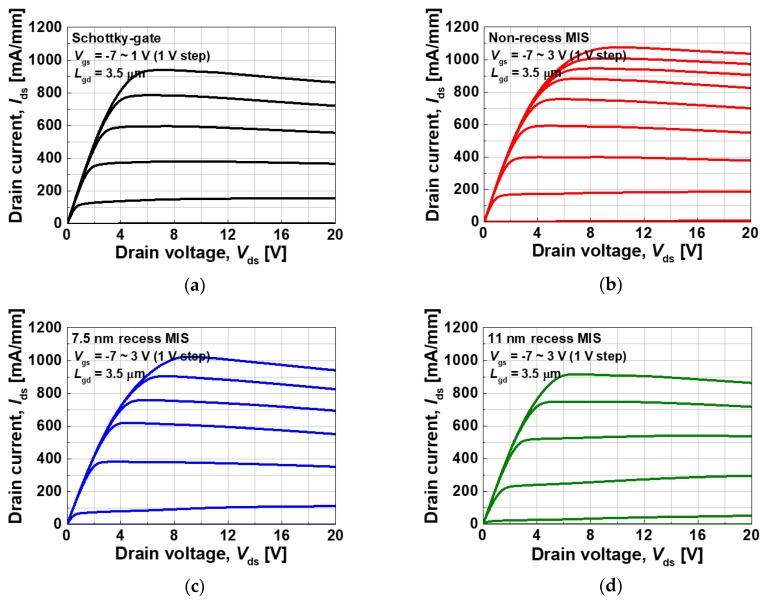
Output characteristics of the fabricated devices; (**a**) Schottky-gate, (**b**) non-recess MIS, (**c**) 7.5 nm recess MIS, and (**d**) 11 nm recess MIS structures.

**Figure 5 materials-13-01538-f005:**
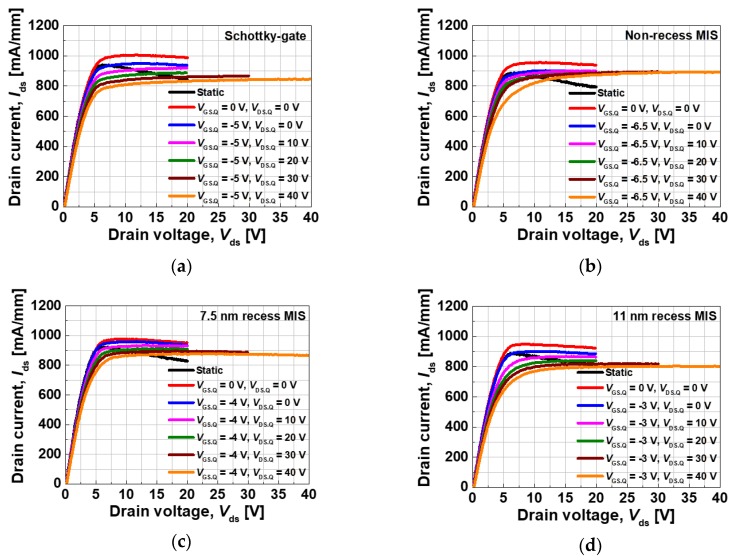
Pulsed I-V characteristics of the fabricated devices; (**a**) Schottky-gate, (**b**) non-recess MIS gate, (**c**) 7.5 nm recess MIS gate, and (**d**) 11 nm recess MIS gate.

**Figure 6 materials-13-01538-f006:**
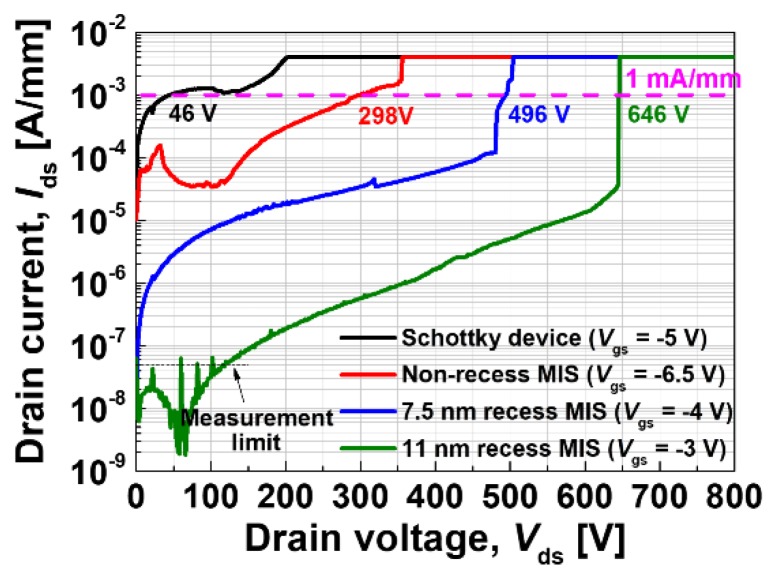
Off-state breakdown characteristics of the fabricated devices.

**Figure 7 materials-13-01538-f007:**
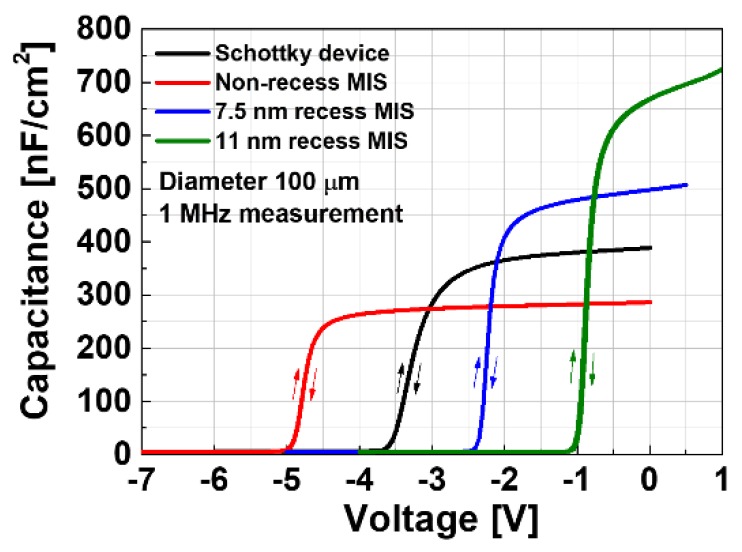
Gate capacitance-voltage characteristics of different gate structures in circular process control monitor patterns.

**Figure 8 materials-13-01538-f008:**
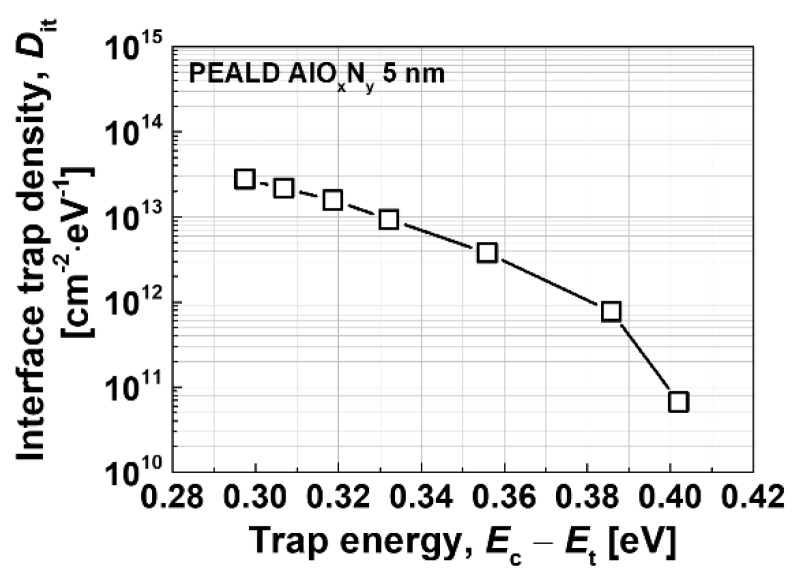
Interface trap density characteristics extracted by a conductance method.

**Figure 9 materials-13-01538-f009:**
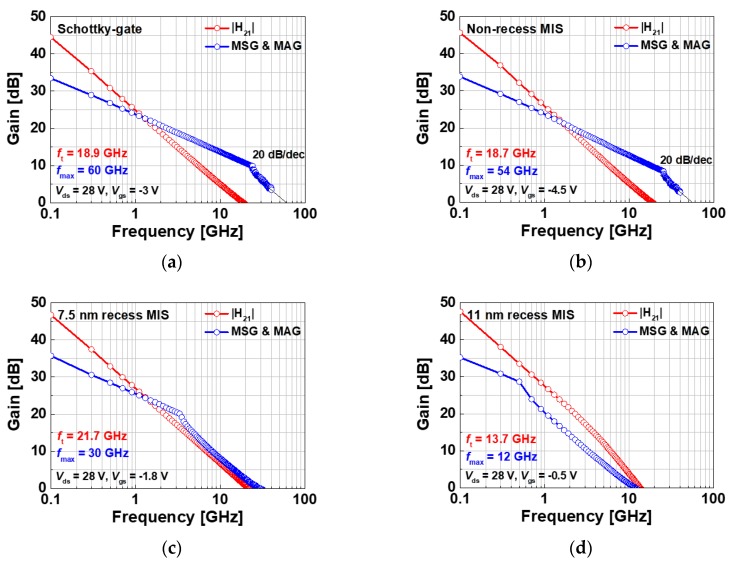
Small-signal characteristics of the fabricated devices; (**a**) Schottky-gate, (**b**) non-recess MIS, (**c**) 7.5 nm recess MIS, and (**d**) 11 nm recess MIS devices.

**Figure 10 materials-13-01538-f010:**
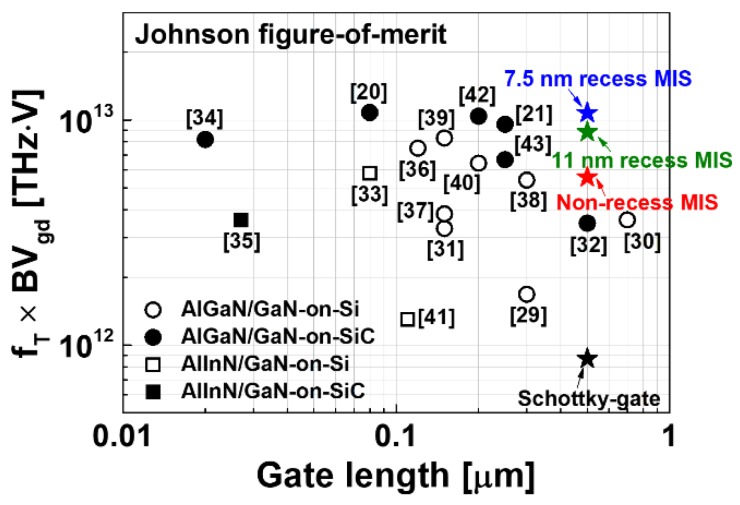
J-FOM as a function of gate length reported for GaN-based HEMTs.

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
