# Peer review of "Effects of Recessed-Gate Structure on AlGaN/GaN-on-SiC MIS-HEMTs with Thin AlO_x_N_y_ MIS Gate"

_materials, 2020, doi:10.3390/ma13071538_

Round 1

Reviewer 1 Report

The authors report the performance of recessed gate MIS-HEMTs with AlON.  please see the following comments. 

  1. MIS-HEMTs with AlON have been reported in the literature. the authors should clarify the motivation and explain the differences between the reported literature. 
  2. how did the author to confirm the thickness of AlON and recessed depth? its better to provide the TEM
  3. what is the k value for AlON? and atom percentage in AlON?
  4. In order to fully evaluate the AlON, the gate breakdown voltage charactersitics should be provided. 
  5.   Fig. 2b shows the two curves of gm for each device. is this the forward-reverse sweep?  the authors should provide the detailed hysteresis analysis
  6. How come the gm is increases as the deep recessed? please explain it. 
  7. please explain the reasons that Fig. 4 shows the results till different VD?
  8. Fig. 6 is not the evidence to prove the quality of AlON since the sweeping voltage is low. the CV sweep should be characterized up to high voltage till the 2nd peak in order to fully understand the trapping effects below the dielectric and to extract the dielectric constant. 
  9. in order to further understand the quality of AlON in the fabricated devices, the freq. C/G-V measurement and Dit analysis should be conducted. 
  10. what is the gate bias applied during the off-state breakdown in Fig. 5? 
  11. please explain the reason that AlON can enhance the breakdown voltage. I suggest use TCAD to verify the results. 
  12. Fe-doped buffer is used to improve the RF performance but C-doped buffer is used to improve the breakdown voltage. It is surprised to see the Fe-doped buffer can have up to 650V breakdown voltage. please explain the reasons. 
  13. Can author provide the vertical buffer breakdown to understand the ideal breakdown voltage?  

Reviewer 2 Report

I believe that this article can be published in its existing form.

Author Response

Thank you for the reviewer’s comments.

Reviewer 3 Report

Interesting paper on GaN Technology HEMTs

Possible improvement concerning adding power measurements could be really important

Some figure readability should be improved, for example Figures 4 have too small legends and too many traces

Fig 6 report capacitance in nF/cm**2: typically, HEMT capacitance is provided not for unit area but fot unit gate length for example pF/mm. Some explanation is required in this context.

Author Response

Thank you for the kind and valuable points. We revised the legends in Fig. 4 (Fig. 5 in the revised manuscript) with enlarged texts.

The C-V characteristics were measured for a circular MIS pattern with a diameter 100 μm. Therefore, the capacitance per unit area are shown in the plot in the manuscript. Since the measurements were carried out using the circular pattern, we think that it is better to keep the unit of nF/cm2.

Round 2

Reviewer 1 Report

Thanks to address the comments.